# Investigation of Energy Cost of Data Compression Algorithms in WSN for IoT Applications

**DOI:** 10.3390/s22197685

**Published:** 2022-10-10

**Authors:** Mukesh Mishra, Gourab Sen Gupta, Xiang Gui

**Affiliations:** Department of Mechanical and Electrical Engineering, School of Food and Advanced Technology, Massey University, Palmerston North 4442, New Zealand

**Keywords:** data compression, RLE, adaptive huffman encoding, H-RLEAHE, IoT

## Abstract

The exponential growth in remote sensing, coupled with advancements in integrated circuits (IC) design and fabrication technology for communication, has prompted the progress of Wireless Sensor Networks (WSN). WSN comprises of sensor nodes and hubs fit for detecting, processing, and communicating remotely. Sensor nodes have limited resources such as memory, energy and computation capabilities restricting their ability to process large volume of data that is generated. Compressing the data before transmission will help alleviate the problem. Many data compression methods have been proposed but mainly for image processing and a vast majority of them are not pertinent on sensor nodes because of memory impediment, energy utilization and handling speed. To overcome this issue, authors in this research have chosen Run Length Encoding (RLE) and Adaptive Huffman Encoding (AHE) data compression techniques as they can be executed on sensor nodes. Both RLE and AHE are capable of balancing compression ratio and energy utilization. In this paper, a hybrid method comprising RLE and AHE, named as H-RLEAHE, is proposed and further investigated for sensor nodes. In order to verify the efficacy of the data compression algorithms, simulations were run, and the results compared with the compression techniques employing RLE, AHE, H-RLEAHE, and without the use of any compression approach for five distinct scenarios. The results demonstrate the RLE’s efficiency, as it surpasses alternative data compression methods in terms of energy efficiency, network speed, packet delivery rate, and residual energy throughout all iterations.

## 1. Introduction and Motivation

The 21st century has been characterised by dramatic shifts in the ways that technology, commerce, and social patterns are organised. The fourth industrial revolution, often known as Industry 4.0, is the result of the trend toward automation and the subsequent reduction in human involvement in production across most sectors [1]. Wireless sensor networks (WSN) and the Internet of Things will play crucial roles in the Fourth Industrial Revolution (IR 4.0 or 4IR). Since IoT devices can move, share, and exchange data without human interaction [2], it enables high flexibility and ease of implementation in a variety of applications. Wireless sensor networks have a limited lifespan due to the significant impact power consumption has on their performance [3]. Energy-efficient media access control and routing protocols [4] are only a couple of the ideas put up to address this problem. Long-term environmental monitoring is a primary goal of a large number of wireless sensor network (WSN) applications. As a result, saving battery power for sensor nodes is an important consideration. Sensor nodes have two main ways to save energy. The use of node redundancy can be a solution to this problem, as it allows a subset of sensor nodes to stay active while the others are put to sleep to save energy. All of the monitored areas must be covered by the subset of active sensor nodes, and the network must remain connected. Furthermore, these sensors must ensure that the network performs as effectively as it does when all the sensors are engaged. We may extend the network’s lifespan by switching between distinct subsets of sensor nodes that are active at the same time. However, such sleep-active techniques may not be implemented if node redundancy is not available (for example, due to network deployment [5,6] or sensor breakdown [7]). Sensor nodes utilise a lot of energy when transmitting data, therefore a possible approach is to decrease the amount of data that is delivered. When sensor nodes are required to send their sensing data to sinks on a regular basis for an extended period, a solution such as this could be extremely helpful [8]. Data from sensing devices must be compressed to save traffic on the network. The data compression system is one of the methods recommended for reducing the amount of data sent through wireless networks. Reduced inter-node communication in wireless sensor networks is a result of using this strategy. Figure 1 illustrates the three kinds of data compression schemes: lossy, lossless, and recoverability of data [9]. Energy, memory, and CPU resources of sensor nodes are extremely restricted; hence, we must choose an efficient and straightforward data compression approach. In order to overcome these concerns, we adapt a lossless data compression approach for use in a wireless sensor network.

The term “lossless compression” refers to a compression method in which, following the execution of the decompression operation, it is possible to recover data that are identical to those obtained before the execution of the compression operation [10]. Sensors in commercial nodes benefit more from lossy compression methods than lossless ones because of the higher compression ratio and lower computational cost that they provide [11]. An accurate and efficient data processing system was developed in [12] to extend the life of clustered WSNs using data prediction, compression, and recovery. The primary objective of this effort is to reduce the cost burden of communication while ensuring the precision of data processing and prediction. The authors of [13] provide a novel architecture that includes a unique combination of data prediction, compression, and recovery in their work. Huffman coding is one of the representative examples for traditional and lifetime maximization cases. When data is compressed using a method known as lossy compression, it is possible that some of the data’s more specific and typically less important properties will be lost as a result of the process. JPEG2000, for example, falls into this family of image and video compression techniques. Finally, from a compressed file in which some data are lost, the recovery tools such as error concealment tools are employed to retrieve the lost data. As shown in Figure 2, there are five broad kinds of WSN data compression algorithms that we will explore in this work.

An overview of these methods may be found here [14]. (1) To compress the data, text data compression strategies are applied to the sensor data in string-based compression approaches. (2) To handle sensing data, image-based compression approaches first hierarchically arrange WSNs before adapting the concept from image compression techniques. (3) The Slepian-Wolf theorem is extended by distributed source coding techniques to encode and decode numerous correlated data streams independently at sensor nodes. (4) A minimal number of randomised and nonadaptive linear projection samples is used by compressed sensing techniques for data compression. (5) Some sensor nodes are responsible for merging data from other sensor nodes in the network as part of data aggregation procedures.

In [15], the authors analyses some of the difficulties that might arise when running a parallel compression method on a CPU/GPU hybrid platform at the secondary sink node of a large-scale wireless sensor network (L-SWSN). It uses the matrix matching principle to dynamically split the compressed data into several dictionary strings and pre-read strings along the vertical and horizontal axes of the GPU’s various blocks, allowing for the simultaneous construction of many matrices.

The simulated approach reported in [16] yields reduced Root Mean Square Error (RMSE) values and larger R2 values (higher coefficient of determination) for varying compression ratios. This research presents an innovative method for enhancing network performance by utilizing Distributed Source Coding and Efficient Energy Consumption while still enabling the delivery of fundamental routing services with reduced traffic delay, end-to-end latency, and total energy consumption. Experiments on their suggested approach show that it performs well, consumes little energy, and can handle a large amount of data [17]. For taking advantage of this spatial relationship, Ref. [18] introduces a joint sparsity-based compressive sensing approach. Their method uses Bayesian inference to create a probabilistic model of the signals, and then they apply a belief propagation algorithm as a decoding technique to restore the original sparse signal. As a result, this research suggests a method for maximizing the usefulness of available sensors by consolidating and compressing data while maintaining its original quality. The primary goal, then, is to minimize this redundant data by eliminating some of the data packets and maintaining just the most crucial ones for the reconstruction. A packet of sensor data was lost during data aggregation, resulting in slower transmission and fewer packet collisions across the cellular connection. The redundancy of storing aggregated data is reduced by data compression, reducing both storage space and transmission bandwidth for use by wireless sensor nodes. As a result of these findings, the lifespan of the network has been greatly extended. The raw data is also considered to be of high quality [19].

The main motivation behind this research is to investigate power consumed during data compression and transmission. Hence, data compression techniques such as RLE and AHE and a hybrid of these techniques are investigated and implemented in different WSN scenarios to achieve higher compression ratio with less power consumption. RLE relies on correlation at the data sources to achieve compression, but AHE may guarantee a greater compression ratio even when the data sources are unknown. The primary issue with RLE is that compression outcomes are affected by the data source. As both the sender and the receiver start out in the communication process in the dark regarding the source sequence statistics, adaptive Huffman coding results in a longer compression time. The H-RLEAHE starts with the compression using RLE technique based on the statistics of the data sources and compresses the data based on that. Further, the compressed data is given as input to the AHE algorithm. The compressed data is communicated to the BS, where it is decompressed using the decompression algorithm.

The outline of the papers is as follows: Section 2 presents the related work for selecting the techniques. Section 3 discusses the basic data compression techniques. Section 4 details the hybrid model for data compression. Section 5 establishes the performance measures and analysis of different data compression algorithms namely RLE, AHE, H-RLEAHE and H-AHERLE. Section 6 discusses the network setup. Section 7 compares the results of different data compression models with RLE, AHE, Hybrid-RLEAHE (H-RLEAHE) and un-compressed data. This paper concludes the entire work with possible avenues of future research.

## 2. Related Work

Wireless sensors are increasingly being used for a variety of new tasks requiring vision, surveillance, object detection, tracking, and geolocation [20]. Long-term environmental monitoring is the primary function of WSN applications, whereas sensor nodes are often powered by batteries. This necessitates that batteries be conserved in order to extend the life of the sensors [21]. The majority of energy consumption in sensor nodes occurs during transmission. Data compression is one way to economise on transmission power. The development of data compression algorithms has become crucial in a variety of applications, particularly in multimedia. One of the most challenging aspects of developing large-scale wireless sensor networks that have real-world applications is coming up with techniques that will enable the network to function over extended periods of time using just the minimum amount of energy that can be stored in or acquired by individual wireless sensor nodes. Since sensor node data transmission is a major drain on the network’s energy reserves, methods for reducing the quantity of data sent between nodes are highly sought after. As a result of this study, network data transfer has been reduced by compressing data locally before it is transferred. In spite of the fact that the topic of data compression has been around for a considerable amount of time, the vast majority of the known methodologies are unable to be immediately transferred to wireless sensor nodes because to the restricted hardware resources, in particular programmed and data memory. Even though compression techniques could be implemented on current wireless sensor nodes, doing so would leave little room for other processes such as sensing and transmission. As a result, these nodes would be less likely to enter deep sleep states, preventing them from realising the energy savings that inspired them to choose a compression strategy. There has been a proliferation of new data compression techniques in recent years for WSNs. Many of these methods can achieve high compression ratios with minimal computing expense, and they can also correlate data collected by sensor nodes.

Among the most popular compression techniques are run-length encoding (RLE), Huffman encoding, Golomb-rice encoding, Lempel-ziv-welch (LZW), and wavelet compression [22]. Text, images, video, and audio are just some of the forms of data that sensors may capture [23]. It is possible that various compression techniques will need to be applied to various forms of telemetry data. Kattan et al. [24] combined LZW and Flate algorithms with JPEG coding and discrete Fourier transforms (DFTs) for textual data compression. For the telemetry data generated by hyperspectral sensors mounted on a satellite, two nearly lossless compression techniques have been presented [25]. Using combined source and channel coding, David et al. [26] developed a distributed compression architecture. Both quantized and correlated side information are used to reduce the amount of inter-node communication required for compression in this method. Qian et al. [27] present an architecture for distributed matched source-channel communication and an algorithm for reconstructing noisy random projections from the data. A similar approach can be found in [28], which uses a gossip communication system. Power-distortion-latency trade-offs exist, despite the fact that universality is stated. In addition, there is no consideration given to the correlation that exists between the data. It was proposed by Logeswaran et al. [29] and Rong et al. [30] that a dis-tributed wavelet analysis architecture be developed that does not rely on grid regularity. How to decide on the optimal path for compression is unclear, and the spatial relationship has not been well investigated. Sensor data from satellite launch vehicles is compressed using a two-stage Lempel-Ziv lossless data compression technique [31]. Launch vehicle data is compressed using a modified version of the Rice compression method [32], resulting in a 2:1 compression ratio. A high compression ratio can be achieved by combining different data compression techniques [33]. It is also critical to implement data compression correctly in hardware in order to improve compression performance [34,35]. To balance hardware costs with a feasible compression ratio, Kao et al. [36] proposed a modularized strategy. Using a two-stage hardware design, Hashempour et al. [37] predicted an increase in compression and decompression rates.

However, in most cases, the objective of deploying a WSN is to monitor a specific occurrence that is of interest. An algorithmic approach that employs both Run Length Encoding (RLE) and basic Huffman encoding to produce compression ratios that outperform current state-of-the art techniques has been suggested in this study. 

Large-scale wireless sensor networks (WSNs) that can be used in everyday life head one of the biggest challenges in making mechanisms that allow the network to run for long periods of time despite the fact that wireless sensor nodes can only store or gather a limited amount of energy [38]. Since sensor node data transmission is a major drain on the network’s energy reserves, approaches to limit the quantity of data sent between nodes are of particular importance. Compressing data locally before transmitting it is the primary goal of this study to minimise network traffic.

Though the field of data compression has been around for a long time, most existing techniques can’t be simply translated to wireless sensor nodes because of the restricted hardware resources, notably programme and data memory [39]. Even though current wireless sensor nodes might run many of the time-consuming compression techniques, this would leave the nodes with limited resources for other functions such as sensing and communication. Using a compression approach meant that these nodes would be less likely to go into deep sleep, which is essential for maximising energy efficiency. WSN-specific data compression algorithms have been presented recently. High compression ratios can be achieved by using algorithms that are computationally cheap, however many of these approaches rely on correlation of data collected by sensor nodes. 

## 3. Data Compression Techniques

The following section elaborates the basics of the constituent algorithms, i.e., RLE and AHE.

### 3.1. RLE (Run Length Encoding)

Run Length Encoding (RLE) is a frequently used compression method. This algorithm’s basic premise is laid forth in [40]. In the case that a data item d appears in the input stream *n* times in a row, we replace each occurrence n with a single pair of data items (n,d). The RLE method [41] is shown graphically in Figure 3. However, the findings of RLE are dependent on the data source because it is based on the same sequential input stream.

### 3.2. Adaptive Huffman Encoding

Huffman coding involves an analysis of the source sequence’s probability. In the case that this information is not easily accessible, the Huffman encoding process transforms into a two-step procedure: in the first step, the statistics are gathered, and in the second step, the source is encoded. Adaptive algorithms based on statistics of previously encountered symbols were created independently to transform this technique into a one-pass approach [35]. To encode the (k + 1)^th^ symbol, we might theoretically recompute the code each time a symbol is delivered by utilizing the Huffman coding process. However, because of the high amount of work required, this strategy would be impractical.

Adaptive Huffman coding employs a technique where neither the transmitter nor the receiver knows the source sequence’s statistics prior to transmission. Nodes in both transmitter and receiver have a weight of 0 and correspond to all symbols that have not yet transmitted (NYT). To keep track of the symbols that have been broadcast, the tree will be updated by adding new nodes to the tree as transmission occurs. Before the transmission begins, the transmitter and receiver agree on a set code for each symbol. In Adaptive Huffman coding, this process becomes a one-pass procedure. NYT are represented by a single node with a weight of zero in both the transmitter and receiver’s trees. Symbols are added to the tree as they are broadcast, and the tree is rearranged using an update mechanism while the transmission continues. Each end of the signal has the same root node as the other. Both the transmitter and the receiver utilize the same method for updates. As a result, the encoding and decoding operations are kept synchronized [42].
(a)Encoding Procedure: Initially, there is just one node in the tree at both the encoder and the decoder, which is the NYT node. As a result, the very first symbol that emerges has a predetermined code-word. When encoding a symbol for the second time, we transmit the code for the NYT node, followed by the fixed code for the symbol that was already agreed upon, unless we are dealing with the very first symbol again. By following the Huffman tree from its root to the NYT node, we may retrieve its corresponding code. This notifies the receiver that the Huffman tree does not yet include a node corresponding to the symbol whose code is about to be received. If a symbol has to be encoded and there is an external node in the tree that corresponds to it, a path from the root node to the external node can be used to produce the symbol’s code.(b)Update Procedure: To perform the update method, the nodes must be arranged in a predetermined order. Using node numbers, this order is maintained.

## 4. Hybrid Model for Run Length Encoding with Adaptive Huffman Encoding

It is evident that the traditional Huffman algorithm necessitates probability distribution in order to generate the Huffman code which may not always be available. In addition, it may not be suitable in scenarios when the probabilities of the input symbols are dynamic. To address the limitations of Huffman coding, researchers have proposed Adaptive Huffman coding technique that uses a novel approach referred to as sibling property to form Huffman tree. As per adaptive Huffman code, the tree initially contains 0-node and maintains a counter for each symbol. As the tree is dynamically created, the generated codes are more effective than Huffman code. As the adaptive Huffman tree is dynamically created, it requires only 1 pass through the input data. Further, as discussed previously, RLE is one of the simplest compression methods that works most effectively for data that contains repeated symbols. However, the most concerning limitation of RLE is that the size of output can be twice to that of input in the worst case. Hence, to address this limitation, we propose a method that combines RLE with AHE as demonstrated in Figure 3.

### 4.1. Hybrid Run Length Encoding with Adaptive Huffman Encoding (H-RLEAHE) Algorithm

The hybrid algorithm works in two phases. First phase involves the encoding through RLE, where the data item d occurs n consecutive times in the input stream and is replaced with the single pair (n,d). The H-RLEAHE is shown graphically in Figure 3. However, the findings of RLE are dependent on the data source because it is based on the same sequential input stream. RLE provides the ST file. In the second phase of the tree, both the transmitter and the receiver employ a single node to represent all unsent symbols NYT with a weight of zero. New nodes will be added to the tree when transmission happens in order to maintain track of the symbols that have been transmitted. To begin transmission, the sender and receiver must first settle on a code for each symbol. NYT node codes are transmitted first, followed by the symbol’s standard code. The symbol is then removed from the NYT list and given its own node. The tree structure of both the transmitter and receiver is same. Both the transmitter and the receiver utilise the same technique for software updates. Encoding and decoding are therefore always carried out at the same time. Nodes must be arranged in a specific order for the update operation to work. Nodes are sequentially numbered to maintain this hierarchy. The root of the tree is given the highest node number, and the NYT node is given the smallest number. Starting with the NYT node and working our way down the tree, the numbers are allocated in ascending order from lower to higher levels. A block is a collection of nodes with the same weight makes up a block.

### 4.2. Hybrid Algorithm

The proposed hybrid algorithm works in two phases: Initially the data is compressed using RLE algorithm, then in the second phase, AHE algorithm is applied on the compressed data received from RLE. The parameters used in the proposed H-RLEAHE algorithm are given in Table 1. Further, the compression technique using the proposed algorithm and decompression is discussed thereafter.

The Algorithm 1 for the H-RLEAHE model is given below:
**Algorithm 1:** H-RLEAHE Compression Algorithm 
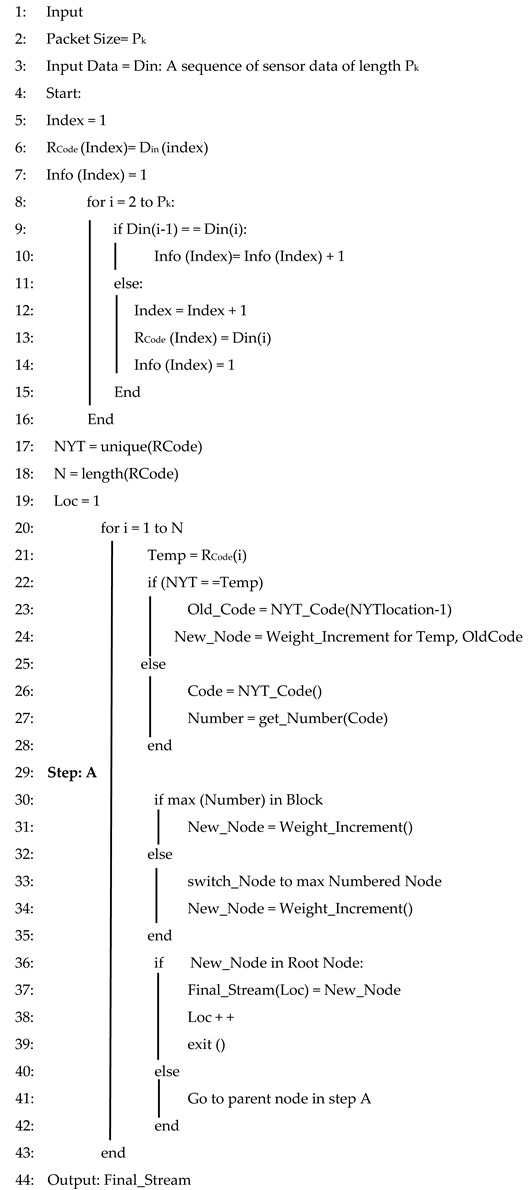


In the H-RLEAHE algorithm, in step 1, data packet is initialized in bits. From step 2 to step 5, RLE algorithm is initialized. During step 6, the *for*-loop is setup and examines to see whether there are any characters that are identical to those in the next index. The count increments to 1 if the characters are identical. If not, the count and character are concatenated. Step 10 checks for unique code and matches and stores it in the location. In step 11, Adaptive Huffman encoding is applied. The Algorithm 2 for decompression is given below.
**Algorithm 2**: Decompression algorithm for H-RLEAHE
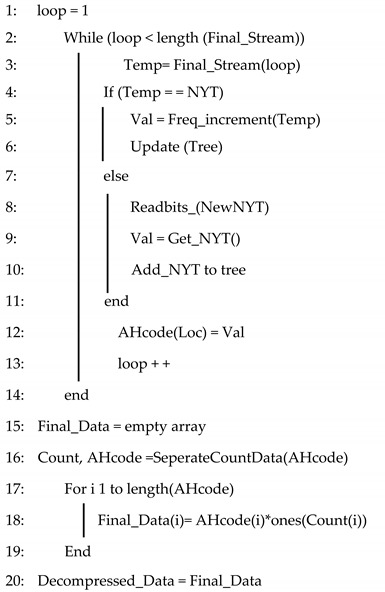


## 5. Performance Measures and Analysis

Analyses of data compression techniques are carried out using the following metrics.
A.Compression Ratio (*CR*): The compression algorithms factor is the ratio of the size of the uncompressed data in bits (*b_uc_*) to the size of the compressed data in bits (*b_c_*) [43].
(1)$$\mathrm{CR}=\frac{\;b_{uc}}{b_{c}}$$
B.Compression Time (*CT*): The time it takes to compress the original data [43].C.Consumption of CPU resources or power used during data compression: An investigation on data compression’s impact on energy use.D.Transmission cost: The amount of energy consumed to transmit compressed data.

### Analysis of RLE, AHE, H-RLEAHE and H-AHERLE Algorithm

In this section we have used real world data [44] to analyze the compression ratio of different algorithms with varying data size. Suppose the length of original data is *L* and the length of compressed data is *K*. We have formulated the energy consumption of different algorithms and used the TIMSP430 [45] microcontroller which has a 16-bit CPU designed for systems with minimal resources. *V_CC_* = 3.3 V, *F_CLK_* = 3.3 MHz, and *I_MSP430_* = 1.85 mA is the current consumption of the TIMSP430 while in active mode. As a result, Equation (2) provides the power consumption per clock cycle of the MSP430 microcontroller. Equation (3) shows how much energy is used to send and receive one bit of data [45,46].
(2)$$E_{CLK\;}=V_{cc}\;\times \;I_{MSP430}/F_{CLK}=1.85\mathrm{nJ}$$
(3)$$S_{bit\;}=V_{cc}\times I_{TX}\;/\;D_{r}=230\mathrm{nJ}$$
where $V_{cc}$ = 3.3 V, $I_{TX}$ = 17.4 mA and $D_{r}$ = 250 kbps

Table 2 shows how basic CPU activity cycles may be utilised to indicate the energy required to transport data while using compression.

Finally, we use the design components to figure out the computational complexity (in number of clock cycles) of different data compression algorithms for compressing the input data using the *EC_algo_* (*L*1, *K*).

Where *EC_algo_* is the energy consumption of algorithms (RLE, AHE, H-RLEAHE an H-AHERLE), *L*1 is the actual input data size and *K* is the compressed data.

We have used actual data input of 1500 bits and the data compressed for difference algorithms is mentioned below. Further, we have evaluated the CPU and transmission cost.
(a)CPU cost for RLE is formulated as in Equation (4)
(4)$$EC_{RLE}=\left(\left(\left(2\times 184\right)+\left(1\times 37\right)\right)\times L1\right)\times 1.85\times 10^{-9}$$where *L*1 is the compressed data of RLE and compressed bits is 70 bits, CPU cost is 0.00018731 J and transmission cost is $S_{bit}\times 70\;\mathrm{bits}$ is 1.61 × 10^−5^ J. RLE algorithim uses two addition operators and one comparison operator.(b)CPU cost for AHE is formulated as in Equation (5)
(5)$$EC_{AHE}=\left(\left(\begin{array}{c} \left(9\times 184\right)+\left(4\times 177\right)+\left(1\times 395\right)+\\ \left(1\times 405\right)+\left(14\times 37\right) \end{array}\right)\times L1\right)\times 1.85\times 10^{-9}$$
where *L*1 is the compressed data of AHE and compressed data is 330 bits, CPU cost is 0.0017029 J and transmission cost is $S_{bit}$ × 30 bits is 7.59 × 10^−5^ J. In Equation (5) there are 9 addition operators 4 subtratctions, 1 multiplication, 1 division and 14 comaprison operators.(c)CPU cost for H-RLEAHE is formulated as in Equation (6)
(6)$$EC_{H-RLEAHE}=\left(\begin{array}{c} \left(\left(2\times 184\right)+\left(1\times 37\right)\right)\times L1+\\ \;\left(\left(9\times 184\right)+\left(4\times 177\right)+\left(1\times 395\right)+\left(1\times 405\right)+\left(14\times 37\right)\right)\times K_{H1} \end{array}\right)\times 1.85\times 10^{-9}$$
where *L*1 is the compressed data of RLE, the $K_{H1}$ is the compressed data of H-RLEAHE and data compressed is 82, CPU cost is 0.00025543 J and transmission cost is $S_{bit}\;\times$ 82 bits is 1.886 × 10^−5^ J. From equation 6 we can observe that there are two addition operators, one compariosn, 23 addition operators, 4 subtractions, 1 division and multiplication and 1 compariosn operators.(d)CPU cost for H-AHERLE is formulated as in Equation (7)
(7)$$EC_{H-AHERLE}=\left(\begin{array}{c} \left(\left(9\times 184\right)+\left(4\times 177\right)+\left(1\times 395\right)+\left(1\times 405\right)+\left(14\times 37\right)\right)\times L1\\ +\left(\left(2\times 184\right)+\left(1\times 37\right)\right)\times K_{H2} \end{array}\right)\times 1.85\times 10^{-9}$$
where *L*1 is the compressed data of AHE, $K_{H2}$ is the compressed data of H-AHERLE and data compressed is 1360, CPU cost is 0.0019502 J and transmission cost is $S_{bit}\;\times \;$1360 bits is 0.0003128 J. Operators used in Equation (7) is 23 additions, 4 subtractions, 1 multiplication and division, and 1 comparison. It can be clearly seen that the transmission cost and CPU cost for RLE is better than the other data compression algorithms.

Hence based on the above equations we have analysed the compression ratio of different algorithms with varying data size from 0 to 1500 bits as shown in Figure 4. It can be clearly seen that the compression ratio of RLE is better than Hybrid-RLEAHE, AHE and Hybrid-AHERLE. For instance, the compression ratio for 1500 bits for RLE is 21.42, H-AHERLE is 18.29, AHE is 4.54 and H-AHERLE is 1.1. Hence from the analysis and compression ratio results we have considered only RLE, AHE and H-RLEAHE in our further studies.

## 6. Network Setup and Scenario Analysis

Different network setup parameters that have been used for investigating data compression algorithm as shown in Table 3. and the assumptions are as follows:

Assumptions

Consider all sensor nodes to be stationary.The study assumes that the data gathered from source IoT nodes is destined for a single BS.SNs that have similar processing and communication capabilities are considered homogeneous. It also takes into account the fact that all SNs have the same initial energy.The x and y coordinates of SNs released randomly are always in the topological region.Applying Euclidean distance, the separation between two near-neighboring SNs is calculated.

### 6.1. Results and Discussion

As indicated in Table 4. the research is further examined in multiple scenarios with varied network area, number of grids/clusters, and total number of nodes. The number of grids varies between 8 and 100, while the total number of nodes spans from 100 to 1250 dependent on the network area. Table 4 provides further information on each of these potential outcomes. The hybrid routing protocol, it is parameter settings and network structure form the basis for different scenarios as reported in [47].

The entire setup is in an area with the base station located at various locations as shown in Table 3. The conventional data compression algorithm RLE and AHE is compared with H-RLEAHE and without compression with the parametric settings given in Table 3 and the grid formation is shown in Table 4. Total packets sent to base station (BS), nodes alive, residual energy in network, packet delivery ratio (PDR), and throughput are used to evaluate performance. In the next sections, we will see how various grid and network settings affect performance.

### 6.2. Case Study 1-1: 2 × 4, 100 Nodes, BS at Center

Figure 5 shows the results of Case Study 1-1: 2 × 4 grids, 100 Nodes, where the base station is placed at the center. Figure 5a depicts the network architecture, which consists of 16 grids with the numbered cluster heads connected by a black line. Figure 5b compares the findings of many different performance metrics and shows that a network with 100 nodes may survive for 118 (AHE), 484 (without compression), 740 (H-RLEAHE), and 982 (RLE) cycles. Compared to RLE, the network lifetime reduces by 87.98%, 50.71% and 24.64%, respectively, for AHE, WC and H-RLEAHE. The increase in the number of rounds for which the network remains alive can be safely attributed to the lesser data to be communicated to the BS as the compression techniques reduce the data significantly. The visualization of average energy in the nodes can be seen from Figure 5c, validating the higher energy in the nodes in the RLE method vis-à-vis other strategies. Further, packet delivery ratio also demonstrates an improvement as shown in Figure 5d. Similarly, Figure 5e demonstrates an improvement in throughput by sending packets up to 982 rounds. The average packets sent to BS/round is also increased significantly as shown in Figure 5f.

### 6.3. Case Study 1-2: 2 × 4, 100 Nodes, BS at Top Edge

This scenario demonstrated in Figure 6 considers placement of BS at top edge. Here, the alive nodes show that with 100 nodes, the network remains alive up to 111 (AHE), 480 (without compression), 733 (H-RLEAHE) and 968 (RLE) rounds, respectively. Compared to RLE, the network lifetime reduces by 88.53%, 50.41% and 24.27%, respectively for AHE, WC and H-RLEAHE. In addition, the throughput reduces for AHE by 88.25%, without compression by 51.28% and for H-RLEAHE it reduces by 25.64% compared to RLE.

### 6.4. Case Study 1-3: 2 × 4, 100 Nodes, BS at Left Edge

The scenario demonstrated in Figure 7 considers placement of BS at left edge. Here, the alive nodes for AHE, without compression and H-RLEAHE techniques reduces by 88.56%, 49.84% and 24.51% rounds, respectively, compared to RLE. In addition, the throughput for AHE, without compression and H-RLEAHE technique is reduced by 88.36%, 50.67% and 25.10%, respectively, compared to RLE.

### 6.5. Case Study 1-4: 2 × 4 Grids, 100 Nodes, BS at Corner

The scenario demonstrated in Figure 8 places BS at a corner. Here, the network lifetime for AHE, without compression and H-RLEAHE techniques reduces by 88.02%, 48.63% and 21.81%, respectively, compared to RLE. In addition, the throughput for AHE, without compression and H-RLEAHE technique is 88.01%, 50.82% and 24.89%, respectively, compared to RLE.

### 6.6. Case Study 2: 4 × 4 Grids, 100 Nodes, BS at Center

Figure 9 shows the results of Case Study 2: 4 × 4 grids, 100 nodes, where the network structure is divided into 16 grids with the different cluster heads deployed at random locations and connected to the BS at the center. The results of alive nodes in Figure 9a shows that with 100 nodes, the network remains alive up to 111 (AHE), 475 (without compression), 719 (H-RLEAHE) and 964 (RLE) rounds, respectively. Compared to RLE, the network lifetime reduces by 88.48%, 50.62% and 25.41%, respectively, for AHE, WC and H-RLEAHE. Further, RLE demonstrates an overall increase in the average energy of the network as shown in Figure 9b. The results in Figure 9c illustrates that packet delivery ratio is higher in RLE as compared to other techniques. Similarly, the total throughput in the network is improved in the RLE as can be visualized in Figure 9d. Finally, packets sent to the BS is also increased for a greater number of rounds in RLE as compared to AHE, no compression and H-RLEAHE, respectively, as shown in Figure 9e.

Comparing Case study 1: 2 × 4 grids, 100 nodes to this, it is clear that while the total number of nodes in both cases is the same, there is a significant distinction in the grid layout. The network performs better in terms of the examined parameters when it is constructed with a 4 × 4 grid. For example, data compression extends network life by 980 rounds as compared to no data compression. Additionally, the network’s topology has shown some improvement in other metrics.

### 6.7. Case Study 3-1: 4 × 4, 200 Nodes, BS at Center

Results from Case Study 3-1: 4 × 4 grids, 200 nodes with the BS at the center of the network are shown in Figure 10. For RLE compression technique the network is alive for 968 rounds compared to 118 rounds, 478 rounds and 722 rounds for AHE, no compression and H-RLEAHE, respectively. This shows alive nodes in Figure 10a for AHE, without compression (WC) and H-RLEAHE reduces by 87.80%, 50.61% and 25.41%, respectively, compared to RLE. Further, the average energy in the network is improved as the compression technique reduces the data leading to lesser transmission of data and thereby more energy in the network Figure 10b. Figure 10c,d also demonstrate similar trends with respect to PDR and throughput. Figure 10e further demonstrates the efficacy of RLE in terms of average packets sent to the BS compared to AHE, no compression and H-RLEAHE techniques, respectively.

To draw the comparison of performance metrics with respect to the case study 2, where the number of grids is same, i.e., 16 but the number of nodes is higher, i.e., 200 nodes, the network remains alive for 968 rounds. Moreover, from Figure 8c–e, minimal improvement in other parameters is witnessed.

### 6.8. Case Study 3-2: 4 × 4, 200 Nodes, BS at Top Edge

The scenario demonstrated in Figure 11 places BS at top edge. Here, the network stays alive for 111 (AHE), 474 (without compression), 722 (H-RLEAHE), and 971 (RLE) rounds. The network lifetime reduces by 88.56%, 51.18% and 25.64% for AHE, WC and H-RLEAHE, respectively, compared to RLE. In addition, the throughput for AHE, without compression and H-RLEAHE technique reduces by 87.95%, 51.69% and 25.95%, respectively, compared to RLE.

### 6.9. Case Study 3-3: 4 × 4, 200 Nodes, BS at Left Edge

The scenario demonstrated in Figure 12 places BS at the left edge. Here, the network lifetime for AHE, without compression and H-RLEAHE techniques reduces by 87.89%, 50.05% and 25.12%, respectively, when compared to RLE. In addition, the throughput for AHE, without compression and H-RLEAHE model is reduced by 88.33%, 51.97% and 25.10%, respectively, compared to RLE.

### 6.10. Case Study 3-4: 4 × 4, 200 Nodes, BS at Corner

The scenario demonstrated in Figure 13 places BS at a corner. Here, the network lifetime for AHE, without compression and H-RLEAHE techniques reduces by 87.80%, 51.01% and 25.80%, respectively, compared to RLE. In addition, the throughput for AHE, without compression and H- RLEAHE techniques reduces by 87.86%, 52.48% and 25.20%, respectively, compared to RLE.

### 6.11. Case Study 4: 10 × 10, 650 Nodes, BS at Center

Figure 14 illustrates the findings of Case Study 4, which was conducted with a grid size of 10 × 10 and 650 nodes. The network topology consisted of 100 grids, and the BS was situated in the center. The results of performance metrics show that the network remains alive for 111 (AHE), 478 (without compression), 722 (H-RLEAHE) and 967 (RLE) rounds, respectively, as shown in Figure 14a. Compared to RLE, the network lifetime reduces by 88.52%, 50.56% and 25.33%, respectively, for AHE, without compression and H-RLEAHE. The average energy in the network is depicted in Figure 14b for the RLE, H-RLEAHE, no compression and AHE technique, respectively. In Figure 14c, the simulation for PDR is demonstrated which shows trivial improvement as compared to previous case studies. Further, in Figure 14d,e RLE demonstrated the increase in residual energy and throughput of the network compared to AHE, no compression and H-RLEAHE.

In case study 4 the performance of the system is still very similar to the performance of the network in the earlier case studies.

### 6.12. Case Study 5: 10 × 10, 1250 Nodes, BS at Center

The findings of Case Study 5 with 10 × 10 grids and 1250 nodes are shown in Figure 15. The network topology consists of 100 grids, and the BS is located at the center. Here the network remains alive for 111 (AHE), 484 (without compression), 726 (H-RLEAHE) and 972 (RLE) rounds, respectively. Compared to RLE, the network lifetime reduces by 88.58%, 50.20% and 25.30%, respectively, for AHE, WC and H-RLEAHE.

### 6.13. Case Study 6: Study of Data Compression with Elliptic Curve Cryptography (ECC) for 2 × 4, 100 Nodes, BS at the Center

As mentioned earlier, the author also investigated network lifetime while including *Elliptic Curve Cryptography* (ECC) that can be considered as an alternative for *public-key cryptography*.

The detailed description of ECC is given by authors in [48,49]. Adding ECC for 2 × 4 and 100 nodes, with BS in the centre, will allow us to fully analyse the data compression technique. This case study illustrates how the compression method performs when the base station is situated on the edge of a 2 × 4 configuration with 100 nodes. The RLE shows a significant increase in different performance indicators in this scenario. Figure 16a shows that the network remains alive for 111 (AHE), 477 (without compression), 719 (H-RLEAHE) and 967 (RLE) rounds, respectively. Compared to RLE, the network lifetime reduces by 88.52%, 50.67% and 25.64%, respectively, for AHE, WC and H-RLEAHE. As shown in Figure 16b–e, RLE achieves an improvement in all the other performance metrics.

## 7. Comparative Analysis with Respect to Alive Nodes

A comparative analysis of alive nodes for the various scenarios is illustrated in Figure 17.

From the graphical representation, it is evident that RLE data compression technique outperforms the other models (Without compression, AHE and H-RLEAHE) for all the scenarios. This strengthens the effectiveness and efficiency of RLE data compression model thus advocating its widespread deployment.

## 8. Conclusions and Future Work

In this paper we have reported the investigative analysis of energy cost of data compression algorithms in WSN for IoT applications. The study includes evaluation of the effectiveness of the established compression techniques, Run Length Encoding (RLE) and Adaptive Huffman Encoding (AHE) for saving energy and thus prolonging lifetime of WSNs. While evaluating energy costs, other metrics such as Packet Delivery Rate (PDR) and throughput have also been considered for the various compression techniques.

While RLE and AHE compression techniques have different strengths and advantages, they also have some drawbacks. This has motivated the authors to embark on proposing and evaluating a hybrid model combining RLE and AHE, named H-RLEAHE. After the compressed data is obtained from RLE, adaptive Huffman encoding is applied to further compress it. The application of RLE ensures that the correlation of data is considered which AHE does not. It is evident from analysis that AHE consumes more energy for compression.

Simulation was carried out for different scenarios wherein the total number of sensor nodes, number of grids and the position of the base station was varied. Compressing the data before transmitting, using any technique, increases the network lifetime when compared to uncompressed data. This is in spite of the extra energy cost of the compression algorithm itself, establishing the efficacy of compression.

Our findings about RLE and AHE conforms with the findings of other researchers who have compared RLE with AHE [50]. RLE requires much smaller memory than AHE. To derive the benefits of each method, in our research we have extended previous research by evaluating the efficacy of combining RLE with AHE. We conclude that for H-RLEAHE the performance metrics are vastly improved over AHE; however, RLE shows the best performance for all the metrics that have been measured. In those scenarios where the sensor data does not have repeatability, AHE performs better that RLE. Hence in applications where data repeatability is less, but the sensor node has sufficient memory, the proposed hybrid model will be most effective.

The modern era of IoT has brought new obstacles for WSNs. Critical information is shared between a wide variety of networked devices, each of which may use a different set of resources that are, in general, limited. The need to meet stringent constraints, such as real-time reaction, high compression ratio, and data transmission efficiency, has motivated designers to create hybrid algorithms that make the most of the devices’ limited resources [51]. This research aims to offer a hybrid data compression approach by fusing Run-Length Encoding and Adaptive Huffman Coding, two well-known loss-less compression methods. If you are trying to cut down on the quantity of data you have to send across a network for medical purposes, this integrated solution is a great option [52].

In addition to efficiency of data transmission, the protection and security of these IoT based sensor applications from malicious adversary are also important. The study is completed by incorporating data encryption and evaluating its implications on energy costs and network lifetime. This is important for practical WSNs where the sensed data needs to be secured before compression and transmission. In terms of security, ECC is studied, and simulation results show that it does not add any significant overheads which can cause the network lifetime to increase. However, if more complex encryption algorithms, such as Elliptic Curve Digital Signature (ECDSA) and Elliptic Curve Diffie-Hellman (ECDH) [53] are used, the effect is likely to be significant. This would form part of future investigative research.

In the future, efforts may broaden towards data aggregation to further enhance the network’s efficiency. Data aggregation involves combining data packets using various bio-inspired routing strategies [54,55]. To do this, the minimum, maximum, and/or mean of sets of data obtained from sensor nodes are manipulated before being sent on to the sink. It is important to use routing algorithms and data compression techniques for effective data aggregation [56].

## Figures and Tables

**Figure 1 sensors-22-07685-f001:**
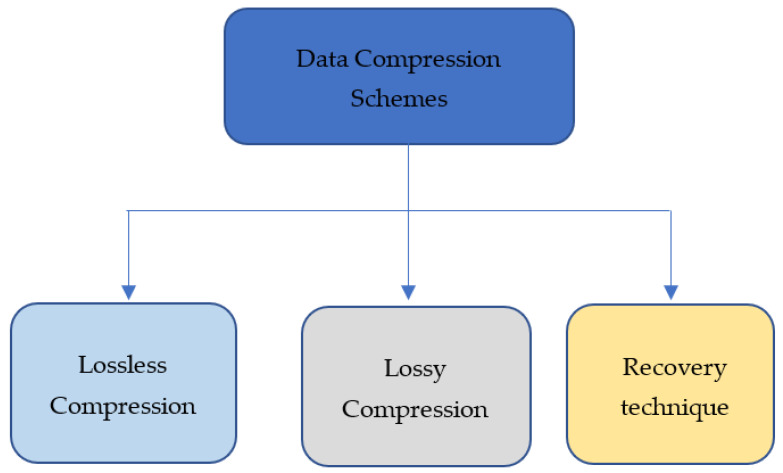
Classification of Data Compression Techniques [9].

**Figure 2 sensors-22-07685-f002:**
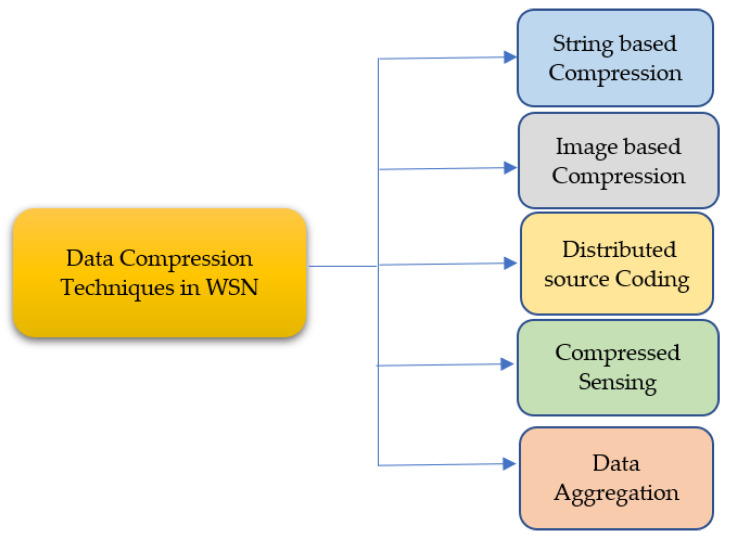
Categorization of Data Compression Techniques in WSN [14].

**Figure 3 sensors-22-07685-f003:**
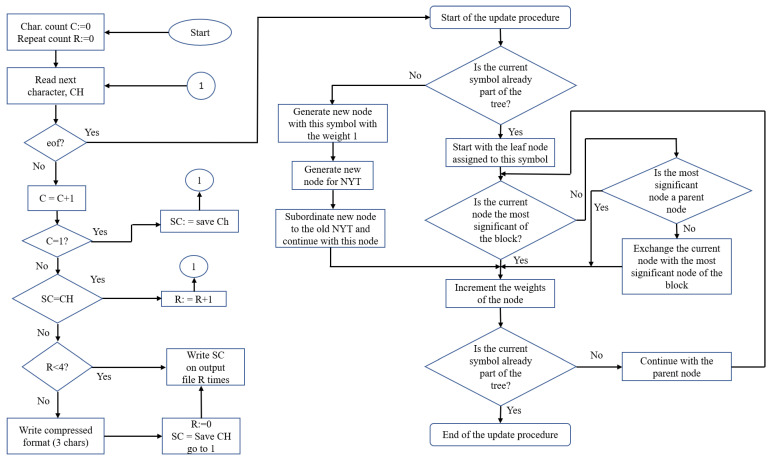
The flowchart of the Hybrid Model.

**Figure 4 sensors-22-07685-f004:**
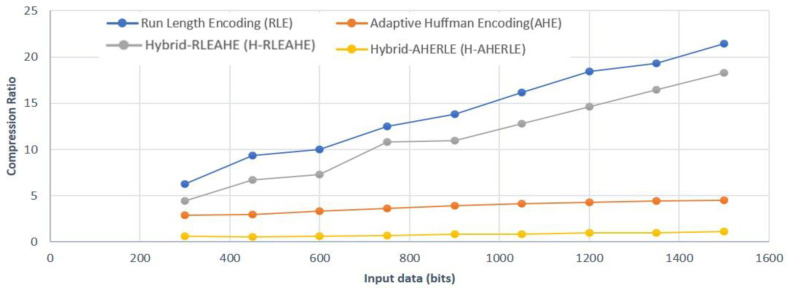
Compression ratio with data varying size in bits.

**Figure 5 sensors-22-07685-f005:**
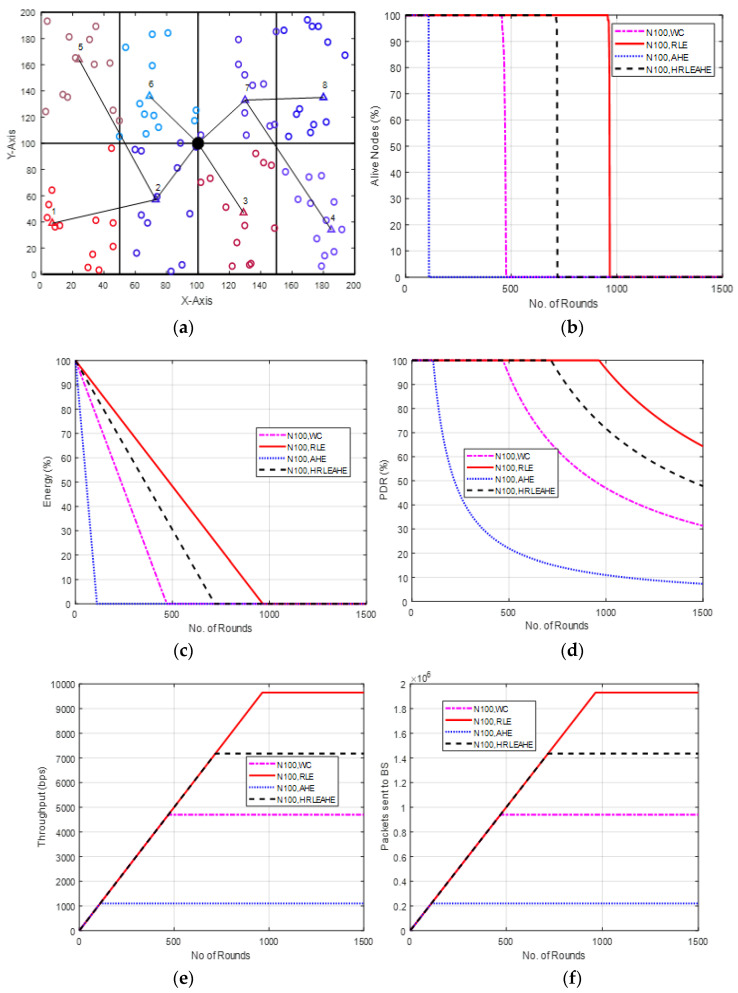
(**a**) Network Structure, (**b**) Alive Node in Case Study 1-1, (**c**) Energy in Case Study 1-1, (**d**) PDR in Case Study 1-1, (**e**) Throughput in Case Study 1-1, (**f**) Packets sent to BS in Case Study 1-1.

**Figure 6 sensors-22-07685-f006:**
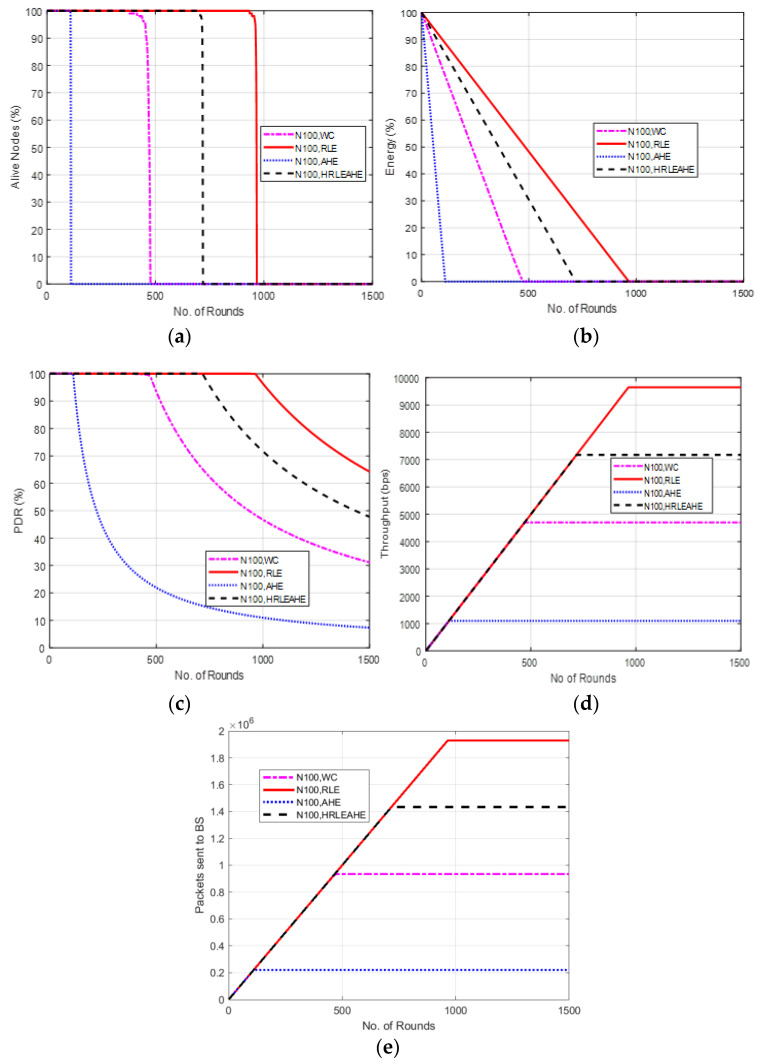
Results of Case Study 1-2: 2 × 4, 100 Nodes, BS at top edge. (**a**) Alive Nodes in Case Study 1-2, (**b**) Energy in Case Study 1-2. (**c**) PDR in Case Study 1-2, (**d**) Throughput in Case Study 1-2, (**e**) Packets sent to BS in Case Study 1-2.

**Figure 7 sensors-22-07685-f007:**
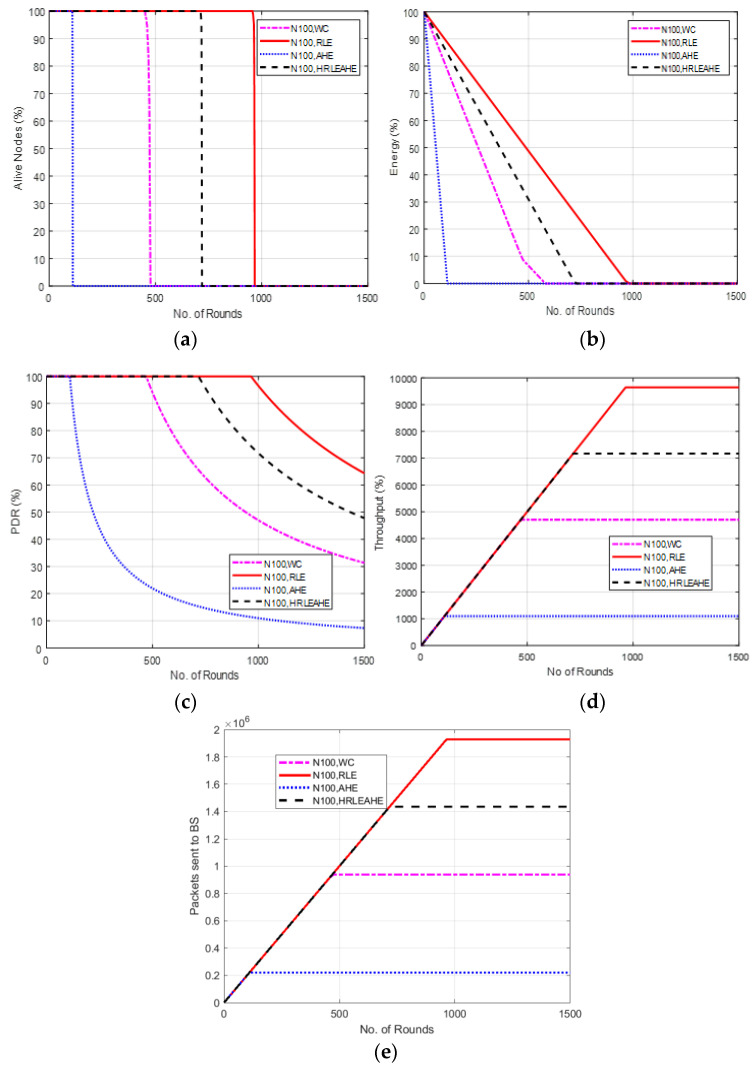
Results of Case Study 1-3: 2 × 4, 100 Nodes, BS at left edge. (**a**) Alive Nodes in Case Study 1-3, (**b**) Energy in Case Study 1-3, (**c**) PDR in Case Study 1-3, (**d**) Throughput in Case Study 1-3, (**e**) Packets sent to BS in Case Study 1-3.

**Figure 8 sensors-22-07685-f008:**
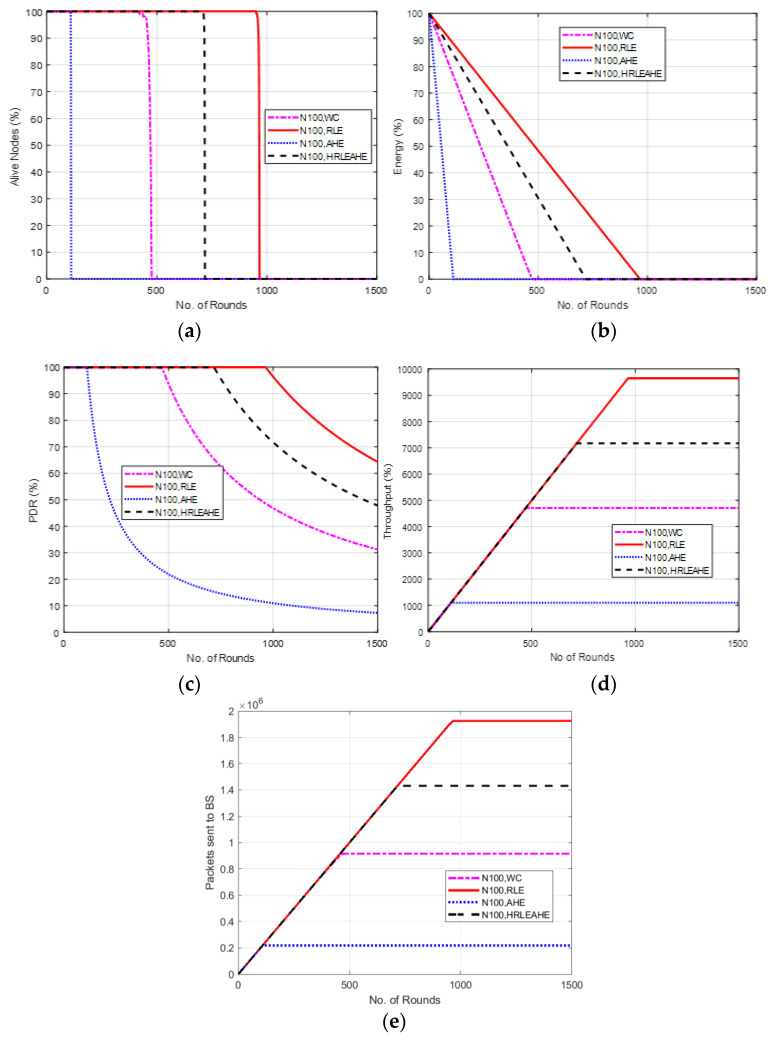
Results of Case Study 1-4: 2 × 4, 100 Nodes, BS at corner. (**a**) Alive Nodes in Case Study 1-4, (**b**) Energy in Case Study 1-4, (**c**) PDR in Case Study 1-4, (**d**) Throughput in Case Study 1-4, (**e**) Packets sent to BS in Case Study 1-4.

**Figure 9 sensors-22-07685-f009:**
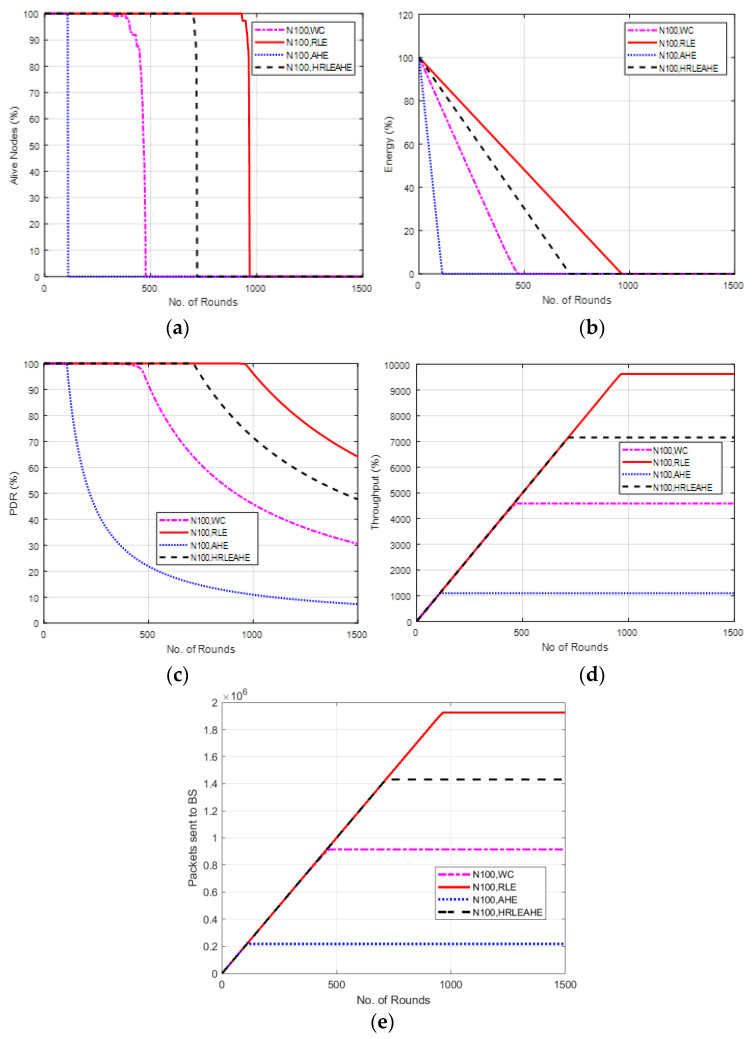
Results of Case Study 2: 4 × 4, 100 Nodes, BS at Center. (**a**) Alive Nodes for Case Study 2, (**b**) Energy for Case Study 2, (**c**) PDR for Case Study 2, (**d**) Throughput for Case Study 2, (**e**) Packets send to BS for Case Study 2.

**Figure 10 sensors-22-07685-f010:**
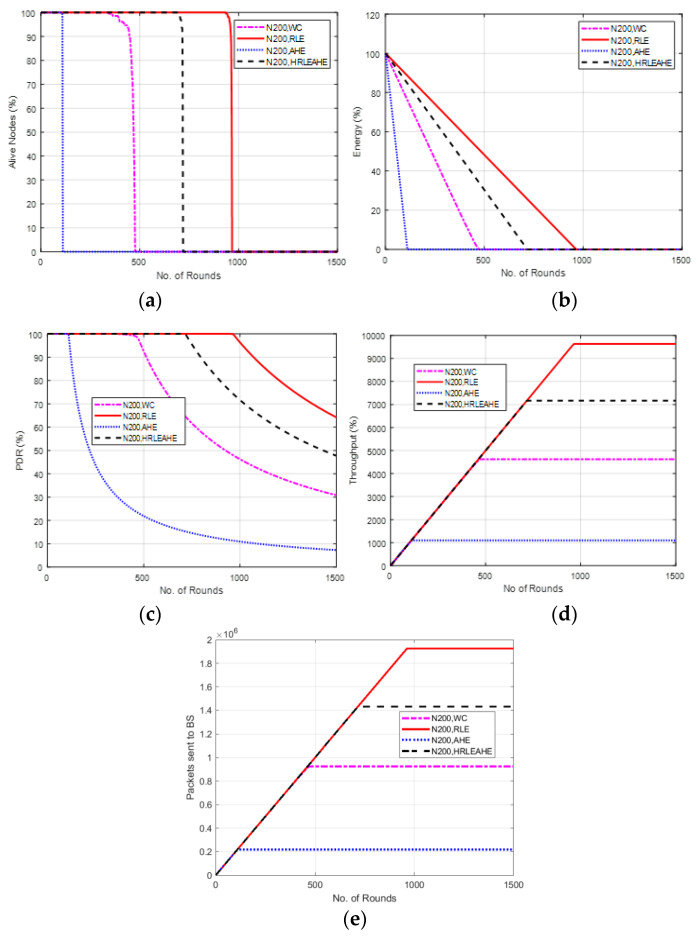
Results of Case Study 3-1: 4 × 4, 200 Nodes, BS at Centre. (**a**) Alive Nodes for Case Study 3-1, (**b**) Energy for Case Study 3-1, (**c**) PDR for Case Study 3-1, (**d**) Throughput for Case Study 3-1, (**e**) Packets sent to BS for Case Study 3-1.

**Figure 11 sensors-22-07685-f011:**
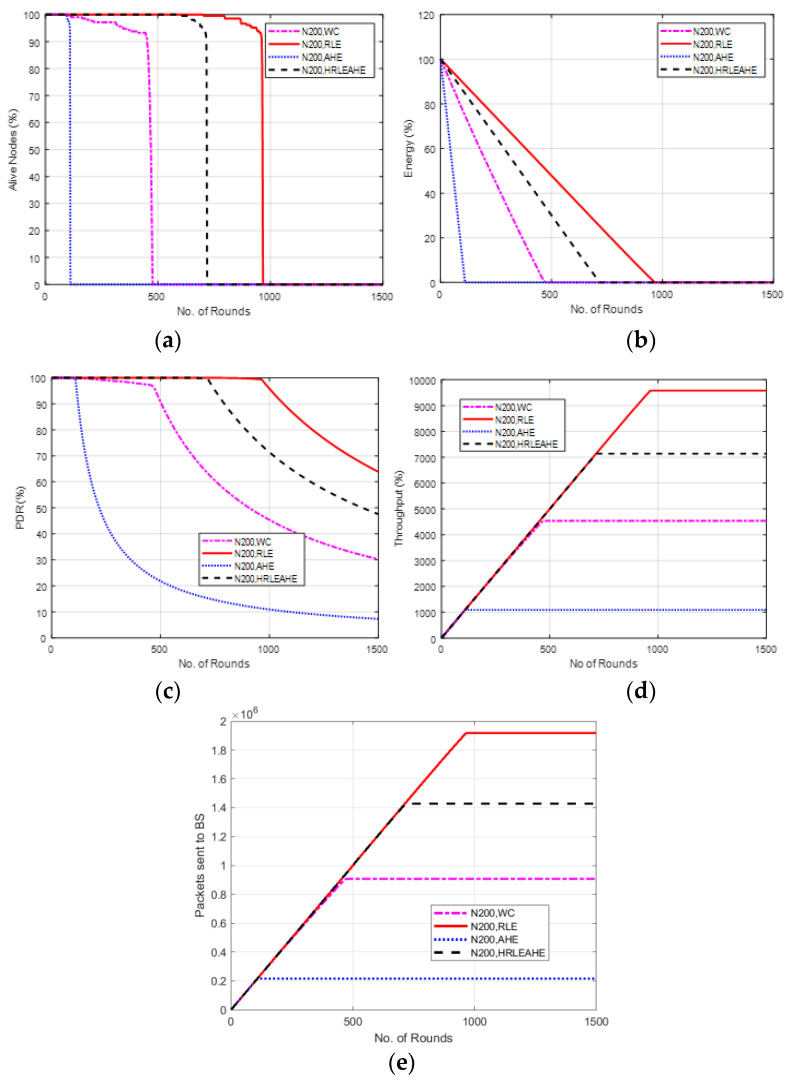
Results of Case Study 3-2: 4 × 4, 200 Nodes, BS at top edge. (**a**) Alive Nodes for Case Study 3-2, (**b**) Energy for Case Study 3-2, (**c**) PDR for Case Study 3-2, (**d**) Throughput for Case Study 3-2, (**e**) Packets sent to BS for Case Study 3-2.

**Figure 12 sensors-22-07685-f012:**
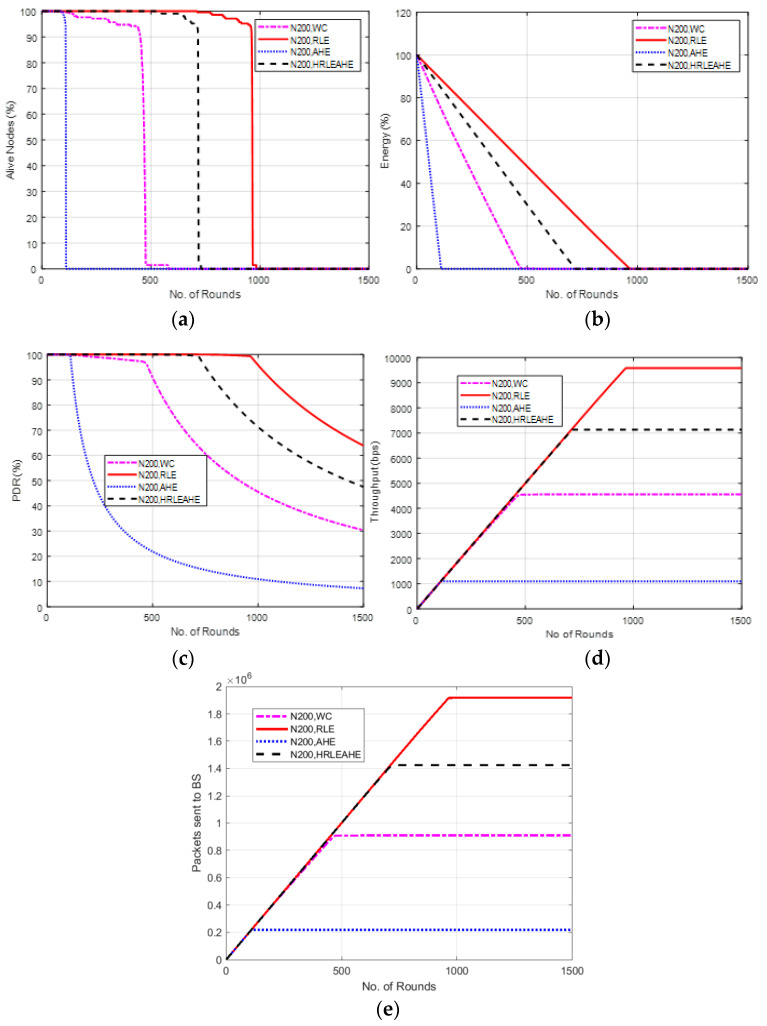
Results of Case Study 3-3: 4 × 4, 200 Nodes, BS at left edge. (**a**) Alive Nodes for Case Study 3-3, (**b**) Energy for Case Study 3-3, (**c**) PDR for Case Study 3-3, (**d**) Throughput for Case Study 3-3, (**e**) Packets sent to BS for Case Study 3-3.

**Figure 13 sensors-22-07685-f013:**
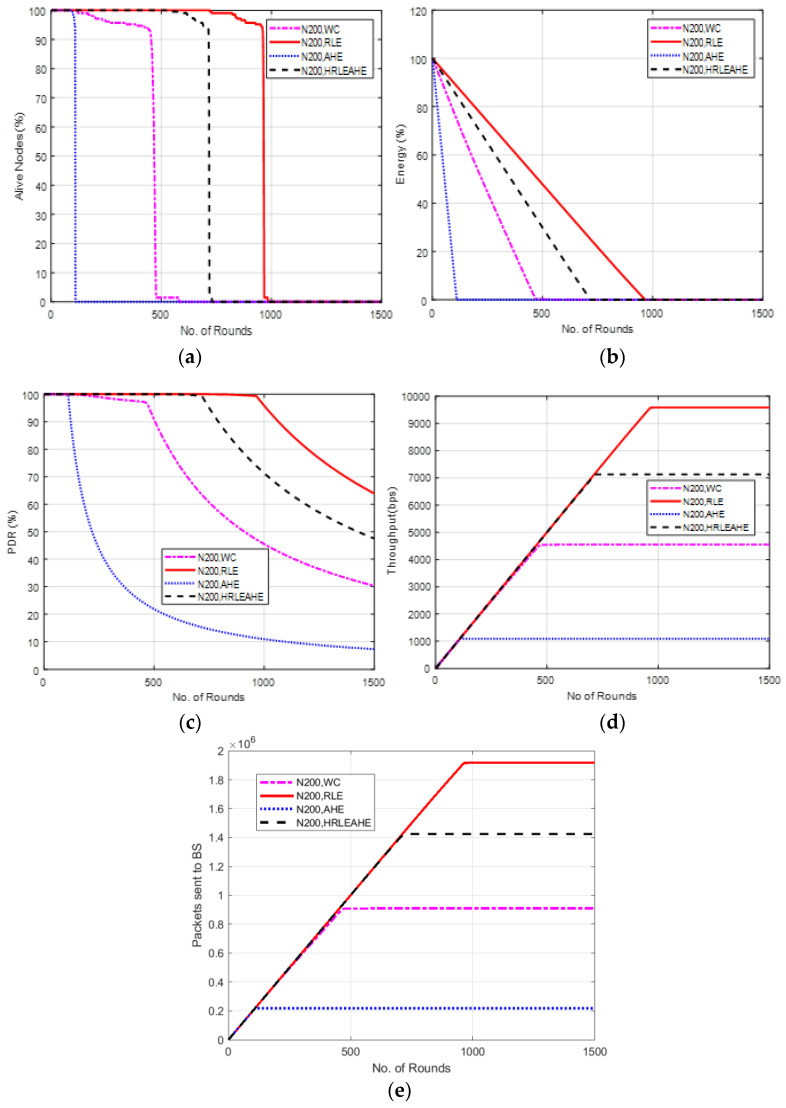
Results of Case Study 3-4: 4 × 4, 200 Nodes, BS at corner. (**a**) Alive Nodes for Case Study 3-4, (**b**) Energy for Case Study 3-4, (**c**) PDR for Case Study 3-4, (**d**) Throughput for Case Study 3-4, (**e**) Packets sent to BS for Case Study 3-4.

**Figure 14 sensors-22-07685-f014:**
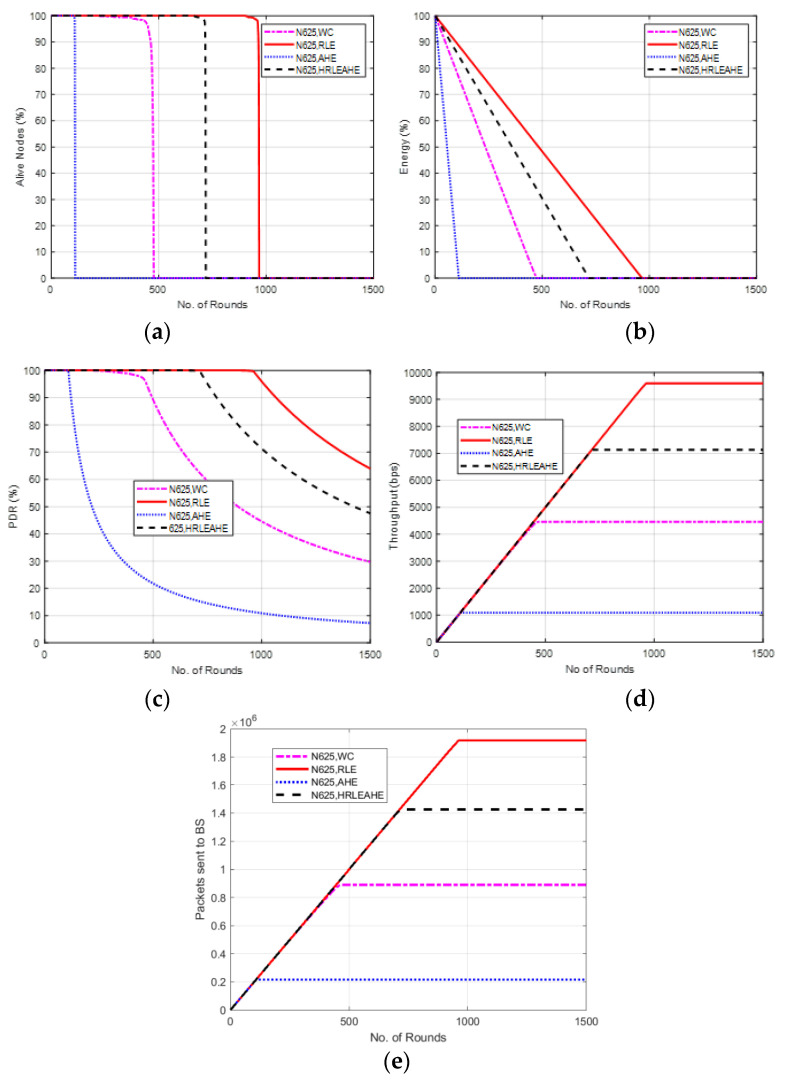
Results of Case Study 4: 10 × 10, 650 nodes. (**a**) Alive Nodes for Case Study, (**b**) Energy for Case Study 4, (**c**) PDR for Case Study 4, (**d**) Throughput for Case Study 4, (**e**) Packets sent to BS for Case Study 4.

**Figure 15 sensors-22-07685-f015:**
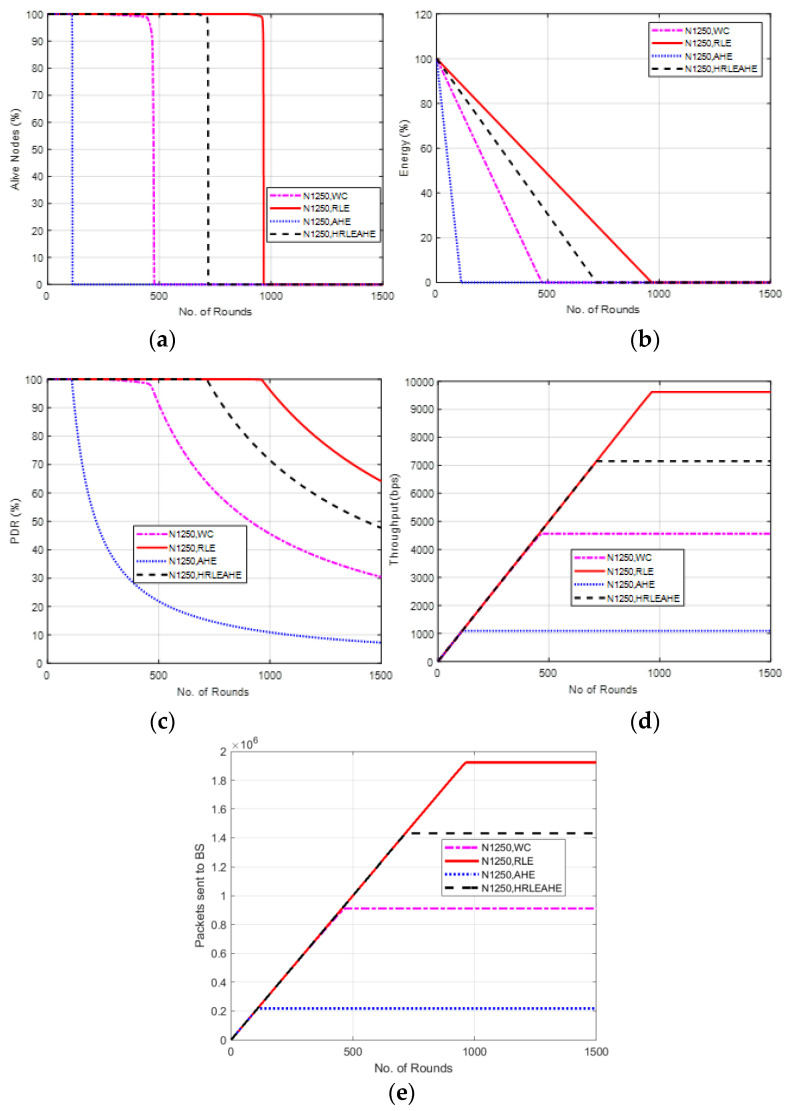
Results of Case Study 5: 10 × 10, 1250 nodes. (**a**) Alive Nodes for Case Study 5, (**b**) Energy for Case Study 5, (**c**) PDR for Case Study 5, (**d**) Throughput for Case Study 5, (**e**) Packets sent to BS for Case Study 5.

**Figure 16 sensors-22-07685-f016:**
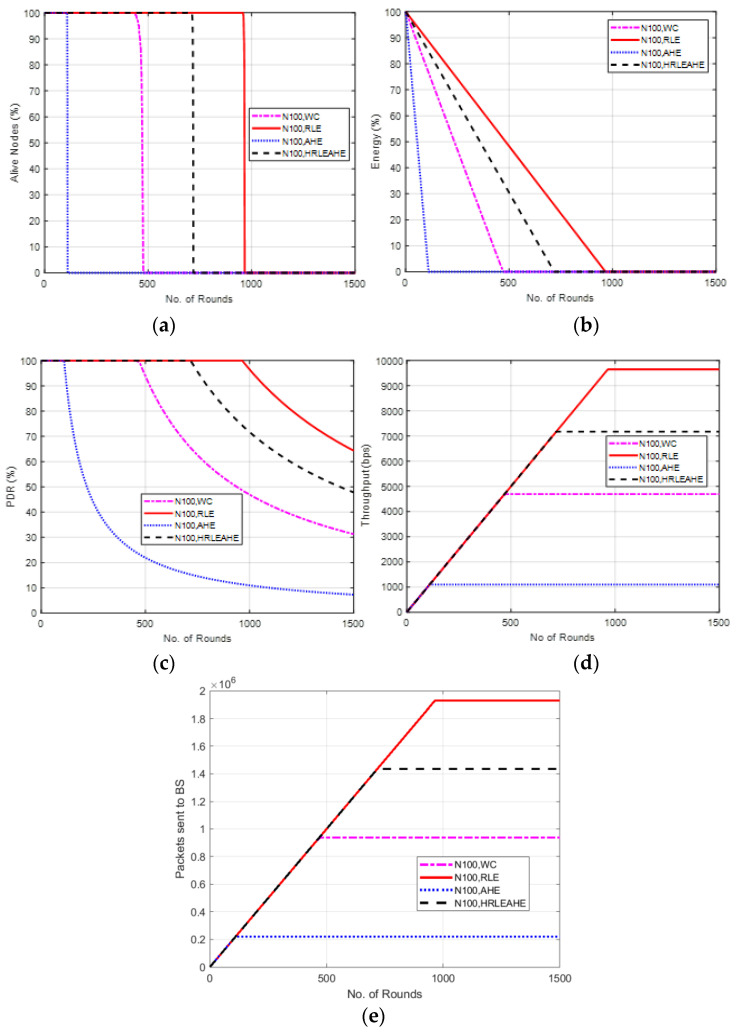
Results of BS at Centre, 2 × 4 100 nodes with ECC. (**a**) Alive Nodes for Case Study 6, (**b**) Energy for Case Study 6, (**c**) PDR for Case Study 6, (**d**) Throughput for Case Study 6, (**e**) Packets sent to BS for Case Study 6.

**Figure 17 sensors-22-07685-f017:**
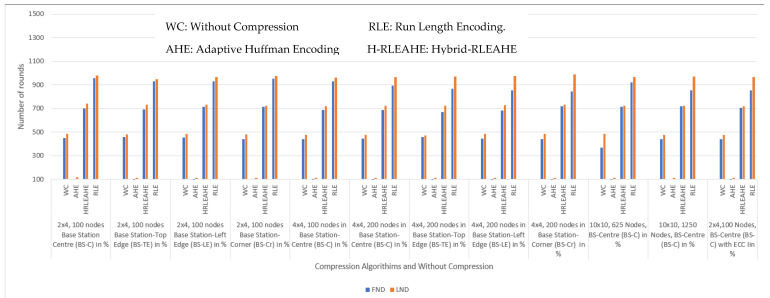
Comparative analysis of various scenarios.

**Table 1 sensors-22-07685-t001:** Parameters that were used in the data compression algorithm.

Parameter	Description
D_in_	A sequence of sensor data
P_k_	Packet size
NYT	Not Yet Transmitted
R_code_	Repeat count in RLE
Length	Length of the stream
Unique	Unique data in the stream
Final stream	Output stream packet
Loc	Location of the Pointer
Info ()	Information about the node

**Table 2 sensors-22-07685-t002:** CPU cycles for the TIMSP430 microocntroller [45].

Operations	Number of CPU Cycles
Addition	184
Subtraction	177
Multiplication	395
Division	405
Comparison	37

**Table 3 sensors-22-07685-t003:** Network parameter setup.

Base station location	2 × 4, 100 nodes (Centre, left edge, top edge and corner) 4 × 4, 100 nodes 4 × 4, 200 nodes (Centre, left edge, top edge and corner) 10 × 10, 625 nodes 10 × 10, 1250 nodes
Number of Rounds	1500 rounds

**Table 4 sensors-22-07685-t004:** Different simulation network configurations [47].

Case Study	Network Area	Number of Grids	Total Number of Nodes
1	100 m × 100 m	2 × 4	100
2	100 m × 100 m	4 × 4	100
3	200 m × 200 m	4 × 4	200
4	500 m × 500 m	10 × 10	650
5	500 m × 500 m	10 × 10	1250

## Data Availability

No new data were created or analyzed in this study. Data sharing is not applicable in this article.

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
