# Peer review of "Investigation of Energy Cost of Data Compression Algorithms in WSN for IoT Applications"

_sensors, 2022, doi:10.3390/s22197685_

Round 1

Reviewer 1 Report

major revision

In this paper, the authors use the RLE and AHE to compress the data for saving the energy cost. There are some issues in this paper

1. It is weird to have the reference citation in the abstract, please revise it

2. Fig. 1 should contain some relevant works as the references, as well as Fig. 2.

3. Motivation should be moved to the introduction section.

4. You dont need figures 3, 4, and 5 as they are the other works not the proposed model in this paper.

5. There is no technical development in Fig. 6, what is the main contribution here while the hybrid model is performed. 

6. Algorithm 1 does not like the pseudo-code format, the authors only modify the program into a simple style, please rewrite it, the same for Algorithm 2

7. Organization of the equations should be re-formatted, they are in a mass. 

8. Fig. 7, you dont need to put numbers on the plot

9. More SOTA models published in recent 2 years should be compared and analyzed in the experiments.

10. Some IoT applications and WSN works should be discussed and cited.

- Suspicious activity detection using deep learning in secure assisted living IoT environments

-Applications of wireless sensor networks and internet of things frameworks in the industry revolution 4.0: A systematic literature review

- Privacy-preserving multiobjective sanitization model in 6G IoT environments

- A sanitization approach to secure shared data in an IoT environment

Author Response

We have addressed all the comments given by reviewer 1

Reviewer 2 Report

In this work, the authors propose a new compression algorithm based on RLE (which depends on the characteristics of the source) and AHE which requires probabilities of symbols and is not adequate in dynamic environments. Their proposal is evaluated in terms of compression ratio, compression time, power consumption and transmission costs.

The authors calculate both energy consumption and computational complexity based on real sensor data and commercial microprocessor which gives a very realistic and useful expressions to evaluate compression schemes.

They compare their hybrid scheme to the original RLE (which still outperforms their proposed scheme in most scenarios) and AHE.

This work is very well written and easy to follow, it is well organized and provides both important results and a clear methodology to evaluate compression schemes in WSNs.

As such, I believe it has enough mearits to be published in its current form.

Author Response

We are thankful for the positive comments given by reviewer 2. 

Reviewer 3 Report

The authors study the problem of energy consumption in devices in wireless sensor networks. They discuss Run Length Encoding (RLE), Adaptive Huffman Encoding (AHE) data compression techniques, and a hybrid of these techniques. Results are based on performance metrics regarding power efficiency, network speed, packet delivery rate, and residual power. The main problem with RLE is that the compression results are affected by the data source. The proposal is very interesting. Here my comments:

1. The model flowchart is confusing, and some boxes do not correspond to its proper use. For example, the conditional. Properly review the correct structures in Figure 3.

2. The quality of the figures is not the same at all. For example, Figures 7, 19(e), and 20.

3. The Related Jobs section is sparse. The authors could complement specific works of currently proposed algorithms, not just make a count of applications. I can recommend some references like:

Rodríguez, A., Del-Valle-Soto, C., & Velázquez, R. (2020). Energy-efficient clustering routing protocol for wireless sensor networks based on yellow saddle goatfish algorithm. Mathematics, 8(9), 1515.

Wang, C. H., Hu, H. S., Zhang, Z. G., Guo, Y. X., & Zhang, J. F. (2022). Distributed energy-efficient clustering routing protocol for wireless sensor networks using affinity propagation and fuzzy logic. Soft Computing, 26(15), 7143-7158.

4. Regarding the Network parameter setup table, it is necessary to complement the metrics with transmission conditions, reception, the algorithm's precision, routing, compression technique, and others.

5. At the end, the authors can make a summary table comparing compression techniques in order to see the comparative gain between them and some others in the literature.

Author Response

We have addressed all the comments given by reviewer 2.

Round 2

Reviewer 1 Report

accept with the current version

Reviewer 3 Report

Thanks to the authors for following the comments. The manuscript would be ready to be published.